# ModelGo: A Pratical Tool for Machine Learning License Analysis

## ABSTRACT

Productionizing machine learning projects is inherently complex, involving a multitude of interconnected components that are assembled like LEGO blocks and evolve throughout development lifecycle. These components encompass software, databases, and models, each subject to various licenses governing their reuse and redistribution. However, existing license analysis approaches for Open Source Software (OSS) are not well-suited for this context. For instance, some projects are licensed without explicitly granting sublicensing rights, or the granted rights can be revoked, potentially exposing their derivatives to legal risks. Indeed, the analysis of licenses in machine learning projects grows significantly more intricate as it involves interactions among diverse types of licenses and licensed materials. To the best of our knowledge, no prior research has delved into the exploration of license conflicts within this domain. In this paper, we introduce ModelGo, a practical tool for auditing potential legal risks in machine learning projects to enhance compliance and fairness. With ModelGo, we present license assessment reports based on 5 use cases with diverse model-reusing scenarios, rendered by real-world machine learning components. Finally, we summarize the reasons behind license conflicts and provide guidelines for minimizing them.

## CCS CONCEPTS

• **Do Not Use This Code → Generate the Correct Terms for Your Paper**; *Generate the Correct Terms for Your Paper*; Generate the Correct Terms for Your Paper; Generate the Correct Terms for Your Paper.

## KEYWORDS

License analysis, AI licensing, model mining

## 1 INTRODUCTION

Over the past decade, the advancement and productization of AI infrastructures have significantly accelerated the proliferation of machine learning (ML) components [25], including AI models [44, 49], software [19, 52], and datasets [13, 47]. Concurrently, the reuse of these components has gained popularity, motivated by concerns about their significant demands on financial and energy resources [48], as well as the widespread recognition of the value advocated by the open-source movement [45]. Unlike code reuse in the OSS field [39], the reuse of AI models follow a distinct scheme. A frequently employed approach for AI models reuse is fine-tuning Pre-Trained Models (PTMs) [17, 49], where PTMs are adapted on a domain-specific dataset, leveraging their robust generalization capabilities.

From a legal perspective, model reuse is generally uncontroversial when its developers or affiliated companies own the copyright for all components. However, data and models often have separate copyright holders in nowadays ML projects [42, 43, 46, 56]. For instance, GPT-2 [42], developed by OpenAI, was trained on 45 million web pages containing personal content and copyrighted materials from third-party platforms like WordPress, GitHub, and IMDb, none of which is owned by OpenAI. These crowdsourced web scraping content [51] typically provides limited usage and distribution rights to users through pre-agreed licenses (e.g., Creative Commons Licenses [8]), which may restrict certain reuse methods like remixing, reproducing, and translating. To prevent legal risk[1], it is essential to ensure that the final ML projects remain compliant with all license conditions associated with the reused components [10, 26, 32].

However, compared to license compliance analysis for OSS, ensuring license compliance in ML projects poses several unique challenges. First, a ML project is not only a combination of software like an OSS project but also composed of datasets and models [17], which may be under different types of licenses (e.g., Free Content Licenses and AI model licenses [9]). Second, ML components often follow more complicated coupling paradigms and nested workflows. For instance, Openjourney [41] is an image generation model derived from StableDiffusion [44], and fine-tuned on images generated by another commercial product, Midjourney [40]. This demonstrates that knowledge can be transferred between models without explicit code integration [55]. Another challenge is improper and ambiguity licensing in ML projects. For example, GPT-2 and BERT [11] are regarded as part of software and then licensed as OSS (e.g., MIT and Apache-2.0). However, ML projects like StableDiffusion and Llama2 [49] tend to apply responsible AI restriction terms for both model and code, using AI model licenses such as OpenRAIL-M [9] and Llama2 Community License [34]. Moreover, to circumvent the limitations of standard OSS licenses, some licensors adopt non-commercial licenses or custom licenses to protect the Intellectual Property (IP) of their models by prohibiting commercial use [22], fine-tuning [30], and reverse engineering [15]. Such ambiguity and the diverse licensing practices within ML projects increase significant legal uncertainty in license compliance analysis. As a result, traditional OSS license analysis approaches [32, 35] only consider replication and linking relationships among software and also lack support for AI model licenses, making them unsuitable for ML projects license analysis.

In this paper, we introduce ModelGo, a tool designed to analyze potential license conflicts, improper license choices, use restrictions and obligations in ML projects that involve nested component reuse procedures. To demonstrate the usefulness of ModelGo, we present 5 use cases constructed using 15 datasets and 11 models from real-world, whose license types cover OSS, free content, and AI model. Our findings show that there exist potential legal risks when reusing components under copyleft, non-public, non-commercial licenses, and point out the need for attention to responsible AI model licenses. The main contributions of our paper are:

- We raise the challenge of license analysis for ML projects and propose ModelGo to assessing it. To the best of our knowledge, our work is the first attempt to deal with this challenge in the ML context.

---

[1] Copyright infringement and privacy lawsuits against OpenAI: 3:23-cv-03199, 3:23-cv-04625, 3:23-cv-03223, 3:23-cv-04557, 3:23-cv-03416, 1:23-cv-08292.

- As part of our work, we introduce a new taxonomy based on the forms of reused components to identify the applicable conditions for various ML reuse mechanisms. This method helps mitigate ambiguity in cases of mismatch between declared license type and actual component type, allowing ModelGo to analyze components under various license types, including OSS, free content, and AI models.
- We provide license compliance reports based on 5 use cases to showcase the effectiveness of our approach. Through our use cases, we offer valuable insights and experiences in achieving compliance in ML projects. Additionally, we also provide license choosing recommendations to minize the risk of non-compliance.

The rest of the paper is organized as follows. Section 2 introduces related studies and the motivations behind this work. Section 3 presents the detailed design, including our proposed taxonomy for bridging AI activities and license language, ML work dependencies structure, and the license analysis workflow of ModelGo. Section 4 provides five case studies and their corresponding findings, and Section 5 concludes this work. **Code will be available once the paper is published.**

## 2 BACKGROUND AND RELATED WORK

In this section, we present the motivations for this work by introducing the background and prior related studies.

### 2.1 Machine Learning Project Licensing

Typically, a ML project is constructed with data, software and models, which are usually governed by different licensing frameworks. To profile current ML licensing, we summary licensing details for ML projects with over 1,000 likes available in Huggingface[2] model repository (See Appendix A.2). Due to a lack of license management in development, we have to manually collect the license information from Huggingface, GitHub, related websites and publications.

**Data Licensing in ML**. Based on our profile, half of ML projects claim their data is licensed in a mixture manner. Additionally, 25% of projects use a single dataset with a standard data license like Creative Commons (CC). The data source of remaining projects (25%) is unknown. Obviously, legal compliance cannot be guaranteed when using data from unknown sources. However, there is also potential risk associated with using datasets under a mixture of licenses or a single license based on follow reasons:

First, the mixture of data sources may involve content under copyleft, non-public, and non-commercial licenses. We investigated the sources of mixture and found that only one dataset, the Pile [13], explicitly removed non-permissive content. Common sources of risk include Wikipedia, arXiv, PubMed and Common Crawl [21] (See Table. 3 for more examples). For instance, sharing derivatives based on non-public licensed content raises suspicion of a license violation, and integrating copyleft content also poses a risk of license incompatibility conflicts. Furthermore, some content sources like IMDb explicitly prohibit data mining in their *Conditions of Use*[3].

Second, the single data license assigned by data collectors may be invalid. In our profile, all datasets with a single license contain

risky data sources. Rajbahadur *et al.* [43] investigated the sources of six public datasets and shown their inherent incompatibility for commercial use. A real case is the copyright infringement lawsuit filed by Getty Images Inc., alleging that Stability AI Ltd. misused Getty Images photos to train its StableDiffusion [44] generative model (1:23-cv-00135). However, the claimed license of training dataset [47] used for StableDiffusion is CC-BY-4.0, which is a permissive license allowing for commercial use. This highlights that ML data licensing is currently irregular and has become a significant factor in legal non-compliance. Although Benjamin et al. [3] have proposed the Montreal Data License (MDL) to foster fair use of data in AI activities, unfortunately, none of the ML projects adopted this license as shown in our profile.

**Software Licensing in ML.** Distinct from OSS projects, only 50% of ML projects release their code with standard OSS licenses. About one-third of ML projects do not declare the code license (but have a model license), which is much higher than in OSS projects [10]. Other projects switch to using AI model or custom licenses to insert additional disclaimers and restrictions related to AI activities, thereby increasing the diversity of licenses in this context. However, given that ML, especially Neural Networks (NNs), is still in its emerging stages, the license dependency chain is shorter compared to OSS projects [4], and most of them use the latest versions of OSS licenses like Apache-2.0 and MIT.

**Model Licensing in ML.** In contrast to software licensing, all ML projects have declared their model licenses. The most popular license is Open Responsible AI License (OpenRAIL) [9], which is a permissive license but includes copyleft-style use-based restrictions governing the use of the model and its derivatives. There are 35% of projects that insist on using unmodified OSS licenses for model licensing, even though these licensing language incurs conceptual ambiguities in the ML context. An interesting finding is that, despite their training data being suspected to contain non-public content, the models are declared as free and open work [21].

**Summary**. ML project licenseing exhibit the following characteristics: 1) Ambiguous, unaccredited and over-permissive license declarations; 2) Emerging RAIL options for model licensing; 3) Unique license dependency structures in ML-specific components reusing. There is a need for new methods to assess ML license compliance.

### 2.2 OSS License Assessment

License analysis for OSS projects has been extensively researched, but it's relatively unexplored in ML context. The research scope and problems of OSS and ML license analysis can be classified into three tiers as shown in Figure 1. For instance, German *et al.* [14] proposed a sentence-based matching tool to identify the license of code. Building on this work, Wu *et al.* [54] further studied inconsistent changes among code clones through provenance analysis. In addition to license identification [24], Vendome et al. [50] proposed a ML-based clustering method to detect license exceptions. These studies mainly deal with copyright issues at the code lines level, located in bottom tier of Figure 1, which can be mapped to similar ML problems: finding the provenance of data sources [43]

---

[2] https://huggingface.co/. Projects in same series but different versions are omitted.
[3] "You may not use data mining, robots, screen scraping, or similar data gathering and extraction tools on this site, .."

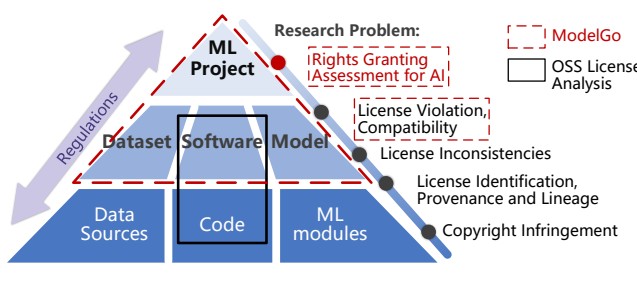

**Figure 1: Research scope and problems of ModelGo compare with traditional OSS license analysis.**

or modules [6]. However, these OSS tools perform software composition analysis through pattern matching or file scanning [35], which are not suitable to datasets and models that typically lack clear provenance and textual licenses.

Shifting the focus to the middle tier, there are some studies that explore license compatibility and violations in software packages [32, 53]. Kapitsaki et al. [26] used Software Package Data Exchange (SPDX) files to detect conflicts in license compatibility (e.g., GPL-2.0 to GPL-3.0). Cui *et al.* [10] directly extracted terms from license texts using Natural Language Processing (NLP) to analyze license conflicts in OSS projects. **However, OSS license analysis works exhibit clear limitations when extended to ML projects.** First, they lack support for dataset and model licenses. For example, RAILs and CCs are not listed in the SPDX. Second, the mixed use of licenses in current ML projects makes it challenging to interpret license conditions across different frameworks. Last, these works only consider code replications and links in their analysis, whereas ML reuse involves a nested and iterative workflow with a more complex dependency structure (e.g., fine-tuning, embedding).

Distinct with previous studies, the research scope of our work is located in top and middle tiers. we propose a practical tool ModelGo to assess potential license violations and non-granting righs errors in ML context. We hope that ModelGo can assist developers in comprehending their obligations and risks when reusing ML components with multiple licenses [1], providing insights for constructing compliant ML systems.

## 3 METHOD

This section is organized around three key questions in ML license analysis: (i) How to determine the applicable conditions in licenses for certain model reuse mechanisms? (ii) How to capture the dependency structure of a ML project? (iii) What types of non-compliance exist in ML projects and how to assess them? We will present our solutions to these questions in the following sections.

### 3.1 Taxonomy for ML License Analysis

Determining the corresponding conditions in licenses is a challenging task for ML projects due to the conceptual ambiguities in existing licensing language and the disorganization in current ML licensing practices. For example, license like CC-BY-ND prohibits the sharing of derivatives of licensed materials. However, its definition of making derivatives is unclear in the ML domain. For instance, should embeddings of a corpus be considered a derivative

work upon that corpus? Unfortunately, even though Creative Commons provides a flow chart to illustrate the trigger conditions of CC licenses in the context of AI activity [7], it raises another question: *Is the output considered protectable copyright subject matter?* The answer depends on how the embedding activity is interpreted, for example, considering it as a translation of the original work can trigger the CC licenses.

MDL advocates the use of a *Top Sheet* to delineate what ML activities are allowed with data [3], but this proposal is rarely implemented in practice (life would be easier if it were widely accepted). Making things more complex, some projects release their models under free content licenses, like LayoutLMv3 model [22], which is licensed under CC-BY-NC-SA-4.0. This disorganization makes it unclear what kinds of ML activities can trigger licenses conditions in different contexts. An ideal and elegant solution would be to encourage licensors to make context-appropriate adaptations in their license agreements or terms of use to clarify the granted rights related to ML activities. However, some ML components may be composed of prior works that are shared under copyleft license templates, which may disallow such relicensing of their derivatives to a new license. Therefore, it is necessary to establish practical rules to bridge AI activities and existing licensing language.

To address the above challenge, we propose a new taxonomy that categorizes all AI activities into four categories based on the forms of their results. There are four categories of AI activities following our taxonomy: Combination, Amalgamation, Distillation, and Generation, which are defined by four forms of their results, respectively: 1) Combination with strong separation; 2) Combination with weak separation; 3) Derivatives from concepts; and 4) Derivatives from data. Correspondingly, we can also categorize the usage behaviors in licensing language into these four categories based on their outcome forms.

We leverage Figure 2 to illustrate this idea. The left side consists of a list of AI activities, many of which pertain to model reusing methods, categorized based on the forms of their results. The middle part is our taxonomy that can classify these AI activities. Following this rule, we can also identify the corresponding terms in natural language license text shown on the right side. For example, Mixture of Experts (MoE) leverages a gating network to ensemble a batch of weak learners [23], which leads to a combination with strong separation and aligns with licensing terms like link, portion, collection, etc. Unlike combination, the results of amalgamation are difficult (or impossible) to separate, corresponding to AI activities such as modification, fine-tuning, model fusion, etc [4]. These unrecoverable revision of original works are corresponding license text like adapt, alter, remix, etc. On the other hand, distillation and generation are derivatives of original works, which means the results will not contain any portion of the original works. These two AI activities are mostly defined in AI model licenses but are not covered by traditional OSS licenses and free content licenses.

By now, we can ascertain the suitable permissions, limitations, and obligations for each AI activity based on the license language, even when the license type is not an exact match. However, its necessarily to emphasize three points. First, our proposed method

---

[4] Whether embeddings constitute a combination with weak separation depends on the specific case. In ModelGo, we classify embeddings as amalgamation if they are created under a content license that treats translation as a form of modification.

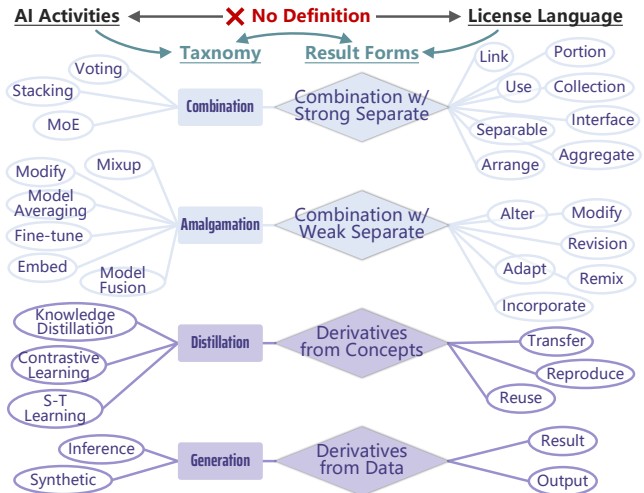

**Figure 2: Our proposed taxonomy bridging AI activities and license language keywords based on their result forms.**

only applies in cases where ambiguities exist in the definition. If the conditions of certain AI activities are explicitly defined in the license, then we should directly follow that. Second, due to the various definitions adopted in different licenses, the final mappings depend on each specific case and may differ from Figure 2. Lastly, one AI activity may trigger multiple license conditions. For example, a fine-tuned model can be seen as a combination with weak separation of the original model, while it can also be viewed as a derivative from fine-tuning data. Therefore, we should design a mechanism to trace these multi-source dependency structures in ML projects, which we will detail in the next section.

## 3.2 Structure of ML Projects

ML projects have unique dependency relationships compared to OSS projects, like the dependencies between generated content and generation model, as well as between training data and trained model. We can summarize these dependencies in ML projects into three categories:

- **Mix-works** be embeded in the new work, either verbatim or in part, in a tangible form. They usually result from direct copying of original components or reusing them through AI activities like combination and amalgamation. These components are embedded into ML projects and must be released with the new work. For example, if we release a new work utilizing Mixture of Experts (MoE), it is equivalent to releasing all weak learners.
- **Sub-works** are similar to mix-works, but the difference is that they are not embedded in the new work. For instance, if we manage to release MoE model along with the data used for training the gating network, then this data will be regarded as the sub-works of MoE model.
- **Aux-works** are components used to build the new work and are either included in it or released with it. For example, the original model used for knowledge distillation.

Figure 3 illustrates the structure of a work constructed by reusing multiple components in ML projects. The final ML project may be constructed through iterative reuse of other works, resulting in

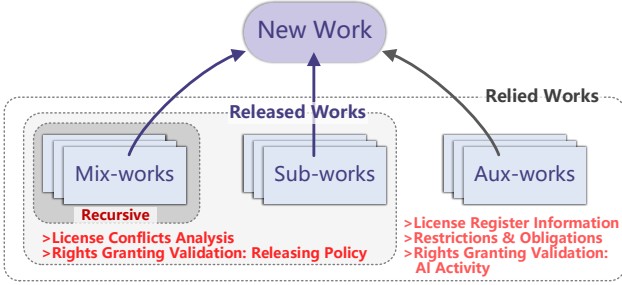

**Figure 3: The proposed structure for capturing work dependencies in ML projects with multiple reused components.**

a ternary dependencies tree for this project. The reason we need this specially-designed tree structure is that works with different dependency types have different license conditions proliferation rules, as illustrated by dotted boxes in Figure 3, which need to be handled separately during subsequent license analysis.

## 3.3 License Analysis in ML Projects

We have outlined all the necessary preparation steps for license analysis in previous sections. Their detailed implementations in ModelGo are as follows.

**Preparation Step 1**: Following our proposed taxonomy, we have manually transcribed the terms in the license text to a standard machine-readable file in YAML format[5]. This file contain following informations for each license:

- Basic license descriptions, including its full name, SPDX short ID, license version, license types (e.g., public domain, permissive, copyleft, proprietary), preferred work types (e.g., software, data, model), and supporting labels such as *disclose code required* and *auto-relicensing applied.*
- Rights granting information, including granted rights and reserved rights as defined by the license text, along with the permitted reusing methods and permitted result forms for redistribution. The prefix of such granting also be noted for cases where the granted rights can be revoked.
- Applicable terms for each AI activity, which contain result forms and relicensability of the activity, corresponding restrictions, and obligations. This item will be marked as *No Defined* if both the activity and the result forms of this activity are not explicitly covered in the license text.

**Preparation Step 2**: To capture the dependency structure of works as shown in Figure 3, we encode the rules of dependencies construction for each AI activity. For example, if we generate embeddings of a corpus using an NN model, then the corpus is considered the sub-work of the generated embeddings, with the activity labeled as *embed*, and the NN model is categorized as the aux-work with the activity labeled as *use*. Furthermore, if the corpus is a collection of smaller corpora, then these smaller corpuses are categorized as the mix-works of the integrated corpus, with the activity labeled as *combine*. By recursively traversing this dependencies tree, we can gather all the dependent works and the activities used to build this ML project.

---
[5] We attempted to use chatGPT to generate this content, but it often behaved unreliably in understanding our taxonomy and produced some stochastic answers [2].

It is important to emphasize a concept in our license analysis approach called *activity proliferation*, which means that the activity performed by a work will recursively proliferate to all its mix-works. In the example of the corpus collection mentioned above, the *embed* calculation performed on the collection will be applied to all the smaller corpuses, triggering their license conditions related to *embed* as well. Similarly, as shown in Figure 3, all rights granting validation and license conflicts analysis of a work should be proliferated to all its mix-works. On the other hand, aux-works are not released with the project, so they are out of the scope of license conflict analysis and rights granting validation for release. In summary, mix-works, sub-works, and aux-works have different scopes in ML license analysis, which is why we need to distinguish between them.

**Analysis Step**: Given the license information and dependencies tree of ML projects, we are ready to analyze the license conflicts within it. ModelGo's license analysis consists of three phases:

*Initial phase*, where we register each component with exact license name, version, type, and format (e.g., raw, binary, SaaS), and then construct their workflows using our predefined reusing functions to capture the dependencies. The release policy should be preset here, and we support personal use, sharing, and selling. Normally, few conditions apply when you only use the work personally, and most license terms limit behaviors like redistribution, sublicensing, and commercial use.

*License determination phase*, where we iteratively derive the eligible new licenses for intermediate reused results. Copyleft proliferation occurs when there is a triggered copyleft license in the relied components. An error will raise if there are other copyleft licenses or if there are components that cannot be relicensed. To condense our analysis results, we prioritize using *Unlicense* for intermediate results once they are relicenseable. After this phase, all components and their derivatives should have a well-determined license name.

*License validation phase*, where we validate the required rights for construct and release this project whether can be granted. The validation also includes compliance with disclosure requirements, such as when a components is in binary format but subject to conditions that require source code disclosure. The releaseability of the final result will be validated upon its mix-works and sub-works, and then an assessment report will be generated.

Table 1 presents the warnings, errors, restrictions, obligations, and notices that can be detected using ModelGo. Table 2 lists the licenses supported by ModelGo, which collectively cover over 96% of licensed models and datasets on Huggingface[6]. In the next section, we will present five case studies based on real ML components.

## 4 CASE STUDY DETAILS

An ideal practice of ModelGo is to assess real-world ML projects and detect their potential license compliance issues. However, this can be challenging in practice due to three present situations:

(1) Prevalent Licensing Disorganization: Many ML projects lack publicly available organized licensing information, making it difficult to ascertain the licenses of individual components.

---

[6] No major changes between different version CCs, so they are all considered as supported. Licenses without clear names and versions are excluded from the calculation. Worth mentioning, our coverage represents only 24.8% and 6.0% of the models and datasets on the entire repository due to the significant number of works without license information.

**Table 1: License warnings, errors, restrictions/obligations, and notices assessed by ModelGo in initial phase, license determination phase and license validation phase.**

| Warning, Error, Restriction, Notice | Description |
|---|---|
| Copyleft / Revocable / No Public Notice | This license or its granted rights are **copyleft / revocable / no public**. |
| License Type Mismatch Warning | License preferred work type is **not compatible** with this work type. |
| License Disclose Self Warning | License requires this work (in binary or SaaS format) to remain **open source** or provide a **readable copy** of the source code. |
| Rights Not Granted Warning | License of this work does **not explicitly grant** you the right to do (...) |
| Rights Not Granted Error | License of this work **cannot grant** you the right to do (...) |
| License Incompatibility Error | Work has a license conflict as it involves **multiple incompatible** licenses. |
| Cannot Relicense Error | Work has a license conflict as it required **relicense** rights not be granted. |
| Cannot Share Error | License **prohibits sharing** of this work. |
| State Changes Restriction | This work must **state changes** according to related license(s). |
| Include License Restriction | This work must retain the **original license file** according to the related license(s). |
| Include Notice Restriction | This work must retain all **notice files** (may contain copyright, patent, trademark and attribution) according to the related license(s). |
| Use Behavioral Restriction | This work must comply with the **use restriction** terms according to related license(s). |
| Runtime Restriction | This work must comply with the **runtime restriction** terms according to related license(s). |

**Table 2: List of licenses (represented by SPDX short IDs) supported by ModelGo, covering over 96% of licensed models and datasets on Huggingface.**

| OSS License (99.8%) | Content License (96.6%) | AI Model License (98.2%) |
|---|---|---|
| Apache-2.0, Unlicense, MIT, AFL-3.0, GPL-3.0, AGPL-3.0, LGPL-3.0, LGPL-2.1, BSD-3-Clause, BSD-3-Clause-Clear, BSD-2-Clause, Artistic-2.0, WTFPL-2.0, OSL-3.0, ECL-2.0 | CC0-1.0, CC-BY-4.0, CC-BY-SA-4.0, CC-BY-NC-4.0, CC-BY-ND-4.0, CC-BY-NC-ND-4.0, CC-BY-NC-SA-4.0, PDDL, C-UDA, LGPL-LR, GFDL | OpenRAIL++, CreativeML-OpenRAIL-M, BigScience-BLOOM-RAIL-1.0, Llama2, OPT-175B, SEER |

(2) Lack of Development Lifecycle Information for ML Reusing: ML reusing often occurs without a clear record, making it hard to trace the origins and licenses of components used.

(3) Non-compliance within Datasets: Crowdsourced datasets often suffer from license non-compliance issues [43], making the licenses (usually permissive) declared by dataset publishers invalid.

Consequently, directly analyzing real-world ML projects can result in uncertainty, over-optimistic results. Therefore, to present more instructive guidelines for assisting developers in understanding the interaction between AI activities and licenses, we have designed five ML scenarios rendered using 15 common data sources and 11 models that cover 5 modalities and 7 tasks, respectively. Table 3 shows the specifications of the involved data sources and models, whose licenses include copyleft, permissive, public domain, and no public license[7]. Furthermore, our case studies can cover all events listed in Table 1, and the their details and findings are provided in the following section.

It's worth noting that, as a license compliance analysis tool, ModelGo's goal is to report potential legal risks in ML projects related to licenses. It is not designed to address legal interpretation issues such as copyrightability of the final work, assessing copyright infringement, or establishing authorship, which typically require verification by a court of law in different regions [20, 31, 36].

---

[7] Some data sources contain crowdsourced content with multiple licenses, and we selected a non-public domain license among them.

**Table 3: Specifications of AI components used in case studies, which include Copyleft License, Permissive License, Public Domain License and Non-Public License.**

| Work Name | License Name | Type | Modality/Usage |
|---|---|---|---|
| Wikipedia | CC-BY-SA-4.0 | Data | Text |
| StackExchange | CC-BY-SA-4.0 | | |
| FreeLaw | CC-BY-ND-4.0 | | |
| arXiv | CC-BY-NC-SA-4.0 | | |
| PubMed | CC-BY-NC-SA-4.0 | | |
| Deep-sequoia | CC-BY-NC-ND-4.0 | | |
| Midjourney Gen | CC-BY-NC-ND-4.0 | | Image |
| Flickr | CC-BY-NC-SA-4.0 | | |
| StockSnap | CC0-1.0 | | |
| Wikimedia | CC-BY-SA-4.0 | | |
| OpenClipart | CC0-1.0 | | |
| ccMixter | CC-BY-NC-4.0 | | Voice |
| Jamendo | CC-BY-NC-ND-4.0 | | |
| Thingverse | CC-BY-NC-SA-4.0 | | 3D model |
| Vimeo | CC-BY-NC-ND-4.0 | | Video |
| Baize | GPL-3.0 | Model | Text Generation |
| BLOOM | BigScience-BLOOM-RAIL-1.0 | | |
| Llama2 | Llama2 Community License | | |
| BigTranslate | GPL-3.0 | | |
| BERT | Apache-2.0 | | Fill-Mask |
| Stable Diffusion | CreativeML-OpenRAIL-M | | Text to Image |
| MaskFormer | CC-BY-NC-4.0 | | Image Segmentation |
| DETR | Apache-2.0 | | |
| Whisper | MIT | | Voice to Text |
| X-Clip | MIT | | Video to Text |
| I2VGen-XL | CC-BY-NC-ND-4.0 | | Image to Video |

## 4.1 CASE I : Corpus Combination

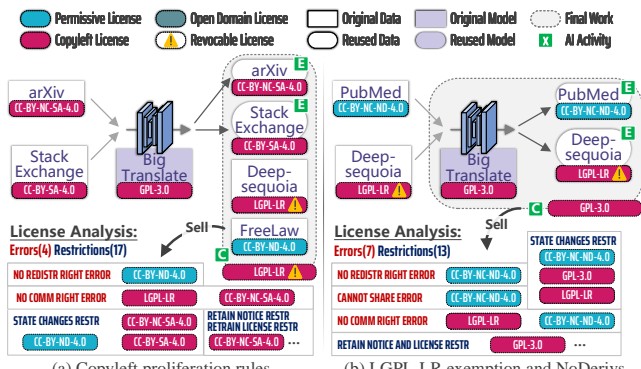

(a) Copyleft proliferation rules  (b) LGPL-LR exemption and NoDerivs

**Figure 4: CASE I: Corpus Combination. AI Activities: E mbed, C ombine.**

Our first case is corpus combination, which is very common in crowdsourced LLM datasets [13, 27, 37]. Additionally, we also consider scenarios where the corpus is extended with the help of translation LLM. As shown in Figure 4 (a), we first translate[8] *arXiv* and *Stack Exchange* using *Big Translate* model, then we combine these translated corpuses with *Deep-sequoia* and *FreeLaw*. This combined corpus is the final work, intended for commercial purposes. Figure 4 (b) depicts a variation in which the final work is a combination of translated corpus and the LLM. Note that, to simplify analysis, we treat these non-public licenses, such as CC-BY-ND-4.0 and CC-BY-NC-ND-4.0, as permissive licenses with limitations on

---
[8] In our cases, we treat translation as a specific form of embedding with a natural language output.

sharing derivatives, as they do not include any copyleft terms. If not specified otherwise, the format of models and datasets is set to raw (i.e., modifiable), while the other supported formats are binary and SaaS. The interpretation of license analysis results is as follows:

Results of CASE I (a) The copyleft conditions about *translation* of the CCs were triggered, which means that the translated corpuses are also covered by the original licenses. As a result, the translated *arXiv* and *Stack Exchange* corpuses remain under the original copyleft CC ShareAlike licenses. However, combining these corpuses with another copyleft-licensed *Deep-sequoia* corpus did not result in the multiple copyleft licenses issue, as the combination with strong separation falls outside the proliferate coverage of LGPL-LR and CC ShareAlike licenses [7]. But, the proliferation extended to the final work and force it to be licensed under LGPL-LR as well. It is important to note that only the effort taken to combine the corpuses is under LGPL-LR, and the licensing action to the final work will not change the inherent licenses of its components.

There are two types of errors according to ModelGo's assessment. The first error arises from the CC-BY-NC-SA-4.0 license of the translated *arXiv*, which doesn't grant the right of commercial use[9]. The second error is caused by the fact that the redistribution rights of final work are not granted to comply with FreeLaw's CC-BY-ND-4.0 license. There are also many restrictions, such as the final work must state the changes compared to the original work and must retain the licenses and notice files of the original works. In addition, ModelGo also indicates that the granted rights of LGPL-LR are revocable, which poses a potential risk for further redistribution.

Results of CASE I (b) Different from CASE I (a), the final work in CASE I (b) is licensed under another copyleft license GPL-3.0 from *Big Translate*. This is because LGPL-LR has a license proliferation exemption for reused results that are no longer classified as linguistic resources. Consequently, the license of final work is proliferated by GPL-3.0. Additionally, besides the rights not granted error arising from CC-BY-NC-ND-4.0, this non-public license also explicitly prohibits any form of sharing derivatives, resulting in a cannot share error.

> **Findings 1**: To minimize the license violation risk when collecting ML data, avoid using content under non-public or non-commercial licenses, and be cautious about the proliferation scope of GPL-like licenses. Based on our assessment, using CC-licensed content (including CC ShareAlike) carries less risk.

## 4.2 CASE II : Mixture of Experts

In this case study, we consider the MoE scenario, in which we combine two models with a newly trained model using a gating network. There are two variations in this case, each involving different models, training data, release policies (SaaS and sharing), as depicted in Figure 5 (a) and (b), respectively. A real-wrold counterpart could be Wu Dao 2.0, which is a LLM trained using MoE technology with

---
[9] This error also arises from *Deep-sequoia* and *arXiv* (since it is a sub-work of the translated *arXiv*), we will omit this type of redundant in the rest of the case studies.

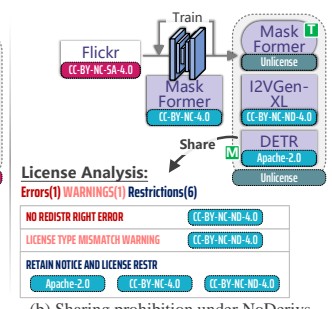

(a) Incompatibility between GPL and RAIL    (b) Sharing prohibition under NoDerivs

**Figure 5: CASE II: Mixture of Experts. AI Activities:** T **rain,** M **oE.**

input from tens of thousands of experts [19]. Additionally, releasing models as a service is commonly observed in commercial AI applications such as chatGPT and Midjourney.

Results of CASE II (a)  There is still significant legal uncertainty regarding whether CC-licensed works can be applied to AI training [7]. Since there is no explicit definition of AI training and corresponding restrictions for resulting models within the license text, we consider training as an undefined activity that falls outside the scope of CC agreements. Therefore, even though the copyleft CC-BY-SA-4.0 license is used for *Wikimedia*, the trained model *BERT* does not trigger the license proliferation conditions and can be relicensed to Unlicense. The final work's license is proliferated to GPL-3.0 from *Baize*, as in CASE I (b).

There is one error in the assessment: the copyleft-style user behavioral restriction claimed in BLOOM-RAIL-1.0 is consider as *non-permissive additional terms*, which can conflict with GPL-3.0. Therefore, an license incompatibility error is reported when we combine *Baize* and *BLOOM* using MoE. The warning is that the final work released as SaaS should remain open source or provide a readable copy of the source code to comply with GPL-3.0. Meanwhile, user behavioral restrictions also apply to the final work, as it is a derivative of *BLOOM* governed by responsible AI conditions [9].

Results of CASE II (b)  In this case study, we replaced experts with CV models. The assessment reveals that the final work cannot be shared, whether modified or not, even for non-commercial purposes, if the project includes CC NoDerivs licenses, as these licenses do not grant redistribution rights to the licensee. This feature is helpful for licensors who intend to prohibit any derivation and commercialization of their models without the need to draft a custom proprietary license. However, this disorganization of ML projects' licensing has a negative effect on the entire ecosystem.

> **Findings 2**: Both OSS and CC licenses lack definitions and corresponding limitations related to model training, leaving freedom to use the trained results. However, RAILs provide comprehensive definitions for AI activities and copyleft-style restrictions, making their derivatives not GPL-compatible.

## 4.3 CASE III : Generation Pipeline

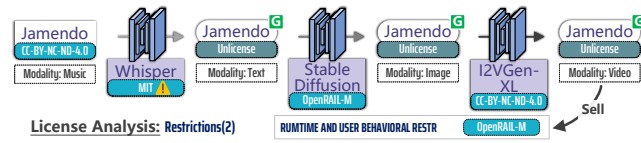

**Figure 6: CASE III: Generation Pipeline. AI Activities:** G **eneration.**

As shown in Table 4, artifact generation has become the most popular application of ML. In this case study, we leverages generative models to produce data for different modalities in a pipeline fashion. The final generated content is released for commercial use.

Results of CASE III  There is still an ambiguity in traditional OSS licenses and free content licenses when it comes to the use of licensed materials for generating artifacts. From the perspective of the license agreement, this AI activity is permitted as long as the *Use* right is granted, and there are also no further claims for the generated content. However, there is one restriction from OpenRAIL-M. The AI model license clearly defines the conditions for AI activities and applies copyleft-style restrictions to its licensed work. Therefore, once AI model licensed components are used in ML projects, all subsequent work should comply with these user behavioral restrictions, which can potentially lead to the final work becoming closed source [16].

> **Findings 3**: Leveraging generative models can bypass the no-sharing conditions of CC NoDerivs licenses and making the generated content almost ungoverned. However, if RAIL-licensed works are involved, the content should comply with their restrictions, potentially leading to further GPL-compatibility issues.

## 4.4 CASE IV : Knowledge Transfer and Fusion

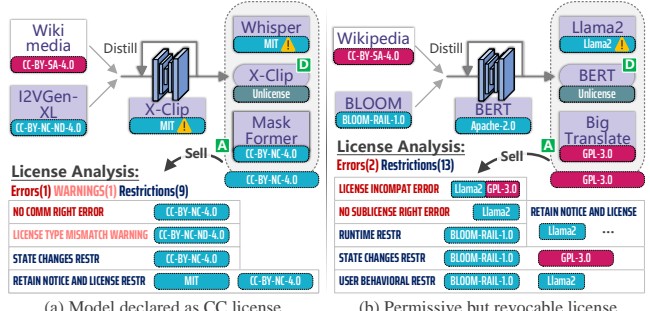

(a) Model declared as CC license    (b) Permissive but revocable license

**Figure 7: CASE IV: Knowledge Transfer and Fusion. AI Activities:** D **istillation,** A **malgamation (e.g., model fusion).**

The knowledge can be transferred or integrated from one model to another without the need for explicit code replication or linking. This is achieved through technologies such as Student-Teacher

Learning [12], Contrastive Learning [29], Federated Learning [33], Model Fusion [28], etc. Traditional OSS licenses expose a loophole regarding these unique reusing methods from ML, and these methods also pose challenges for deep IP protection [38]. With the assistance of ModelGo, we further explore the compliance of these knowledge transfer methods within existing licensing framework.

Results of CASE IV (a)  The knowledge fusion like model averaging and fusion yield a weak separation result from the original work, which can be interpreted as one form of amalgamation. Therefore, the final work should be under a CC-BY-NC-4.0, the same as *Mask Former*. However, the CC licenses do not define the terms for the materials used for distillation, so there is no effect from the copyleft licenses of *Wikimedia* and *I2VGen-XL*.

There is one error in the assessment. Since the modification of a CC NonCommercial licensed work cannot be relicensed according to its conditions, the amalgamated result face a no commercial rights error when commercialized.

Results of CASE IV (b)  This case study assess license compliance towards NLP models. There have two errors all detected from *Llama2*. The first error is the license incompatibility between its use limitations terms and the GPL-3.0. The second error is because the Llama2 license does not grant sublicense rights for further republication, conflicting with the releasing policy. Additionally, the rights granted by the Llama2 license are revocable, posing a potential risk in the final ML project. Furthermore, the final work should also comply with the user behavioral restrictions demanded by BLOOM-RAIL-1.0 and Llama2.

> **Findings 4**: Knowledge transfer is a powerful method to bypass the reproduction prohibition of models. However, model fusion may trigger the terms like remix, incorporate, and adapt, necessitating the reusing procedures to remain in compliance. In addition, the rights may be revocable even if granted by a permissive license.

## 4.5 CASE V : Remix Data

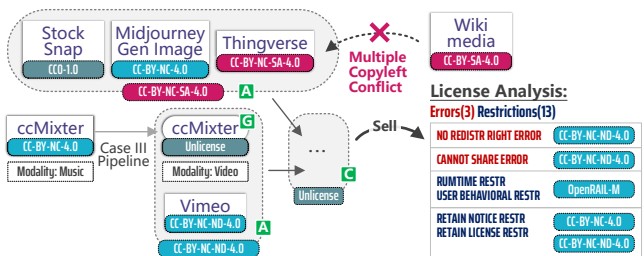

**Figure 8: CASE V: Remix Data. AI Activities:** [G] **generation,** [A] **malgamation,** [C] **ombination.**

Mirroring the CASE IV, this case considers the scenario of data remix and integration, which can arise when using data augmentation methods such as *mixup*[57], SMOTE[5], ADASYN [18], etc. We reuse the generation pipeline depicted in Figure 6 to increase the complexity of the assessment.

Results of CASE V  We first analysis the remix of *StockSnap*, *Midjourney Gen Image* and *Thingverse*. For content under public domain licenses like CC0-1.0, we can freely remix this content without worrying about any conflicts. However, conflicts may arise when remixing content under CC-BY-NC-4.0 and CC-BY-NC-SA-4.0 licenses. As shown in Figure 7 (a), CC-BY-NC-4.0 cannot be relicensed for its remixed result, while CC-BY-NC-SA-4.0 requires performing license proliferation. But the outcome is this remixed work can be relicensed to CC-BY-NC-SA-4.0 because there is a one-way compatibility between CC licenses, as indicated by a supplementary interpretation from Creative Commons[10]. A conflict due to multiple copyleft licenses will arise if we attempt to further remix with *Wikimedia*. Furthermore, there will be a *cannot relicense* issue if we attempt to augment *Wikimedia* and relicense it to a new permissive license to bypass the mentioned conflict.

On the other hand, remixing the generated *ccMixter* and *Vimeo* is governed by CC-BY-NC-ND-4.0, which is responsible for almost all errors and restrictions in the final product. However, we can get rid of these constraints by leveraging the loophole of generative content as shown in CASE III.

> **Findings 5**: Directly remixing raw data should ensure compatibility between licenses, which can be challenging in crowdsourced scenarios. One feasible solution is to exclude all content under copyleft and non-public licenses. An irregular tactic is to exploit the current ambiguity in licensing frameworks regarding generated content.

## 4.6 Summary

Based on the findings from our case studies, we conclude five guidelines to minimize license conflicts and legal risks in ML projects:

(1) Avoid reusing any works under proprietary or unknown licenses, as they may pose a risk of copyright infringement. (2) If you intend to use any ML components under RAILs (or other responsible AI model licenses), avoid including GPL-like licensed works in your projects, and vice versa. (3) Refrain from using any non-public or non-commercial licensed works if you plan to share the project or sell it, respectively. (4) If you're uncertain about compatibility, limit your project to using at most one copyleft license. (5) Ensure that all components are under appropriate licensing frameworks. We provide a flowchart to illustrate this idea in Appendix A.2.

Please note that our guidelines are aimed at minimizing potential risks related to license terms and do not provide legal interpretations as previously mentioned. See our disclaimers in Appendix A.1.

## 5 CONCLUSION

Component reusing is prevalent in today's ML project development lifecycle, yet legal compliance issues are often ignored. Furthermore, it can be challenging for developers to understand elusive license terms and identify the potential risk of license violations. Therefore, given the particularity of ML projects and licensing practices, we propose a practical license analysis tool to analyze their license conflicts. We leverage five case studies to demonstrate the feasibility of our method, and our findings provide constructive guidelines to minimize conflicts.

---

[10] https://wiki.creativecommons.org/wiki/Wiki/cc_license_compatibility

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

# A APPENDIX

## A.1 DISCLAIMER

The content presented in this article is intended for general informational purposes only and should not be construed as legal advice. Any views, opinions, findings, conclusions, or recommendations expressed in this material are the sole responsibility of the author(s) and do not represent the perspectives of any organization or entity.

## A.2 Additional Figure and Table

Figure 9 illustrates the flowchart for minimizing license conflicts in a ML project. Table 4 displays the summary of licensing details for ML projects with over 1K likes on Huggingface. Table 5 presents statistical data related to licenses and their corresponding number of works on Huggingface.

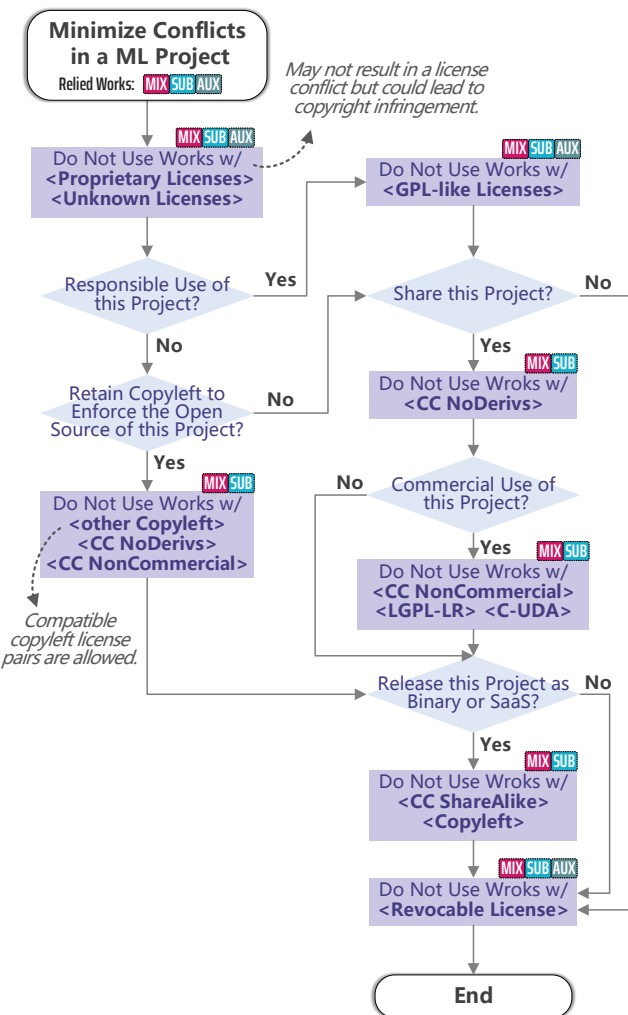

**Figure 9: Flowchart for minimizing license conflicts in ML projects.**

**Table 4: Summary of licensing details for ML projects with over 1K likes on Huggingface (Accessed on October 11, 2023).**

| ML Project | Task | Data License | Software License | Model License | Dataset | Risk Resource |
|---|---|---|---|---|---|---|
| Stable Diffusion v1-5 | Text to Image | CC-BY-4.0 | CreativeML-OpenRAIL-M | CreativeML-OpenRAIL-M | LAION-5B | Common Crawl |
| BLOOM | Text Generation | *Mixture* | *Unknown* | BigScience-BLOOM-RAIL-1.0 | *Crowdsourced* | Common Crawl, Wikipedia, etc. |
| OrangeMixs | Text to Image | *Mixture* | *Unknown* | CreativeML-OpenRAIL-M | *Crowdsourced* | Danbooru |
| ControlNet | Text to Image | *Unknown* | Apache-2.0 | OpenRAIL | *Unknown* | n/a |
| Openjourney | Text to Image | CC-BY-NC-4.0 | *Unknown* | CreativeML-OpenRAIL-M | Midjourney Gen | Midjourney Gen |
| ChatGLM-6B | Text Generation | *Mixture* | Apache-2.0 | *Custom* | the Pile, Wudao, *Crowdsourced* | PubMed, Wikipedia, arXiv, GitHub, etc. |
| Llama2 | Text Generation | *Unknown* | Llama2 Community License | Llama2 Community License | *Unknown* | n/a |
| StarCoder | Text Generation | *Mixture* | Apache-2.0 | BigCode-OpenRAIL-M | The Stack | none |
| Falcon-40B | Text Generation | ODC-By | Apache-2.0 | Apache-2.0 | RefinedWeb | Wikipedia, Reddit, StackOverflow, etc. |
| Waifu Diffusion | Text to Image | *Mixture* | *Unknown* | CreativeML-OpenRAIL-M | *Unknown* | n/a |
| Dolly-v2-12B | Text Generation | CC-BY-SA-3.0&4.0 | MIT | MIT | databricks-dolly -15k, the Pile | PubMed, Wikipedia, arXiv, GitHub, etc. |
| Dreamlike Photoreal | Text to Image | *Unknown* | *Unknown* | *Modified* CreativeML-OpenRAIL-M | *Unknown* | n/a |
| Counterfeit | Text to Image | *Unknow* | *Unknown* | CreativeML-OpenRAIL-M | *Unknown* | n/a |
| GPT-2 | Text Generation | *Mixture* | *Modified* MIT | *Modified* MIT | *Crowdsourced* | WordPress, GitHub, wikiHow, IMDb, etc. |
| GPT-J-6B | Text Generation | *Mixture* | Apache-2.0 | Apache-2.0 | the Pile | PubMed, Wikipedia, arXiv, GitHub, etc. |
| LLaMA-7B | Text Generation | *Mixture* | *Custom* | *Custom* | *Crowdsourced* | GitHub, arXiv, etc. |
| BERT | Fill Mask | *Mixture* | Apache-2.0 | Apache-2.0 | Book Corpus, Wikipedia (en) | Wikipedia (en) |
| Whisper | ASR | *Unknown* | MIT | MIT | *Unknown* | n/a |
| MPT | Text Generation | *Mixture* | Apache-2.0 | Apache-2.0 | *Crowdsourced* | Common Crawl, Wikipedia, etc. |
| Mistral-7B | Text Generation | *Unknow* | Apache-2.0 | Apache-2.0 | *Unknow* | n/a |

**Table 5: List of Huggingface supported licenses and number of works, with ModelGo supported licenses highlighted in BOLD. Note that many works do not explicitly indicate their license version. (Accessed on October 11, 2023).**

| Model Licenses (Total: 355,150) | | Dataset Licenses (Total: 69,277) | |
|---|---|---|---|
| License Name | # of Works | License Name | # of Works |
| **Apache-2.0** | 46,758 | MIT | 5,415 |
| **MIT** | 21,365 | Apache-2.0 | 3,026 |
| OpenRAIL | 17,760 | OpenRAIL | 1,639 |
| **CreativeML-OpenRAIL-M** | 12,059 | **CC-BY-4.0** | 1,355 |
| other | 6,521 | other | 1,257 |
| CC-BY-NC-4.0 | 2,867 | **CC-BY-SA-4.0** | 609 |
| CC-BY-4.0 | 2,676 | AFL-3.0 | 515 |
| **AFL-3.0** | 2,111 | CC | 444 |
| **Llama2** | 1,776 | **CC0-1.0** | 435 |
| CC-BY-NC-SA-4.0 | 1,312 | **CC-BY-NC-4.0** | 385 |
| **GPL-3.0** | 1,080 | **CC-BY-NC-SA-4.0** | 378 |
| CC-BY-SA-4.0 | 959 | CC-BY-SA-3.0 | 377 |
| **OpenRAIL++** | 667 | CreativeML-OpenRAIL-M | 290 |
| CC | 625 | GPL-3.0 | 266 |
| **BigScience-OpenAI-M** | 596 | **CC-BY-NC-ND-4.0** | 190 |
| **Artistic-2.0** | 579 | BigScience-OpenRAIL-M | 114 |
| **BSD-3-Clause** | 525 | CC-BY-3.0 | 94 |
| **BigScience-BLOOM-RAIL-1.0** | 422 | CC-BY-2.0 | 91 |
| **WTFPL** | 331 | Artistic-2.0 | 91 |
| CC-BY-SA-3.0 | 288 | ODC-by | 80 |
| CC0-1.0 | 270 | WTFPL | 80 |
| **BigCode-OpenRAIL-M** | 251 | Unlicense | 68 |
| **AGPL-3.0** | 237 | Llama2 | 63 |
| **Unlicense** | 199 | BSD | 62 |
| CC-BY-NC-ND-4.0 | 194 | GPL | 54 |
| GPL | 173 | **C-UDA** | 49 |
| BSD | 155 | AGPL-3.0 | 46 |
| CC-BY-3.0 | 104 | CC-BY-NC-SA-3.0 | 38 |
| GPL-2.0 | 84 | ODBL | 35 |
| CC-BY-2.0 | 80 | **GFDL** | 34 |
| BSL-1.0 | 75 | BSD-3-Clause | 34 |
| **BSD-2-Clause** | 74 | **CC-BY-ND-4.0** | 32 |
| **LGPL-3.0** | 65 | CC-BY-NC-3.0 | 28 |
| C-UDA | 57 | BigScience-BLOOM-RAIL-1.0 | 28 |
| CC-BY-NC-2.0 | 48 | GPL-2.0 | 26 |
| CC-BY-NC-3.0 | 45 | OpenRAIL++ | 24 |
| **OSL-3.0** | 44 | CC-BY-NC-2.0 | 21 |
| **ECL-2.0** | 35 | BigCode-OpenRAIL-M | 20 |
| PDDL | 35 | **PDDL** | 20 |
| **BSD-3-Clause-Clear** | 28 | BSD-2-Clause | 16 |
| CC-BY-ND-4.0 | 27 | LGPL-3.0 | 15 |
| GFDL | 26 | CDLA-Sharing-1.0 | 14 |
| Ms-PL | 26 | CC-BY-2.5 | 12 |
| Zlib | 25 | Ms-PL | 11 |
| LGPL | 21 | CDLA-Permissive-2.0 | 11 |
| DeepFloyd-IF-License | 19 | CC-BY-NC-SA-2.0 | 10 |
| CC-BY-NC-SA-3.0 | 19 | MPL-2.0 | 10 |
| LGPL-LR | 17 | EUPL-1.1 | 10 |
| MPL-2.0 | 16 | CC-BY-NC-ND-3.0 | 10 |
| ISC | 15 | BSL-1.0 | 10 |
| CC-BY-NC-SA-2.0 | 15 | BSD-3-Clause-Clear | 8 |
| ODBL | 15 | LGPL | 6 |
| CC-BY-2.0 | 14 | ECL-2.0 | 6 |
| CC-BY-NC-ND-3.0 | 14 | OSL-3.0 | 5 |
| ODB-by | 13 | ISC | 5 |
| NCSA | 9 | **LGPL-LR** | 4 |
| EPL-2.0 | 9 | PostgreSQL | 3 |
| EUPL-1.1 | 9 | Zlib | 3 |
| CDLA-Sharing-1.0 | 7 | EPL-2.0 | 2 |
| **LGPL-2.1** | 6 | OFL-1.1 | 2 |
| PostgreSQL | 5 | LGPL-2.1 | 1 |
| LPPL-1.3c | 5 | CDLA-Permissive-1.0 | 1 |
| EPL-1.0 | 4 | CC-BY-2.0 | 1 |
| OFL-1.1 | 3 | NCSA | 1 |
| TII-Falcon-LLM | 2 | DeepFloyd-IF-License | 1 |
| CDLA-Permissive-2.0 | 2 | EPL-1.0 | 1 |
| CDLA-Permissive-1.0 | 2 | LPPL-1.3c | 1 |

