# OpenReview forum: "ModelGo: A Tool for Machine Learning License Analysis"
_ACM.org/TheWebConf/2024/Conference — TheWebConf24 Oral_

### Official Review · Reviewer_wX85 · 2023-11-22

**Novelty:** 6
**Technical Quality:** 5

**Review:**

The paper introduces and discusses ModelGo, a tool designed to analyze potential license conflicts, improper license choices, use restrictions, and obligations in machine learning (ML) projects. It provides insights into the challenges of licensing in ML, such as the intricate interaction of various license types and the need for specialized tools to handle these complexities.

The paper is well-structured, clearly presenting the problem, the methodology, and the application of ModelGo. It successfully articulates the need for such a tool in the ML domain and presents case studies to demonstrate its utility.

The work addresses a relatively unexplored problem in ML license analysis. The authors claim this to be the first attempt to tackle license analysis challenges specifically in the ML context, highlighting its novelty. The significance of this work lies in its practical application to a growing field. As ML projects become more prevalent, the complexity of license management increases. ModelGo offers a solution to navigate this complexity, potentially impacting how ML projects are managed and mitigating legal risks.

Pros

1. Addresses a Critical Gap: The tool addresses the significant issue of license management in ML, which is increasingly important as the field grows

2. Practical Application: ModelGo is designed for practical application, as demonstrated through real-world case studies

3. Methodology: The methodology for license analysis, including the concept of 'activity proliferation', is innovative and tailored to the unique challenges in ML projects.

Cons

1. Complexity: The complexity of the tool might be a barrier for some users, particularly those without a background in legal or license management.

2. Limited Scope: While ModelGo covers a broad range of licenses, it may not encompass all possible licensing scenarios in the rapidly evolving field of ML

3. Dependency on Accurate Data: The effectiveness of ModelGo is contingent on the accurate and comprehensive input of licensing data, which might be a limiting factor.

**Questions:**

1. Scope of Licensing Types: How does ModelGo handle emerging or non-standard licensing types that are increasingly seen in the ML field? Are there plans to regularly update the tool to include new types of licenses?

2. Complexity for Users: Given the complexity of license management in ML, how user-friendly is ModelGo for individuals without a legal or technical background? What measures have been taken to make the tool accessible to a broader audience?

3. Accuracy of License Data: The effectiveness of ModelGo depends on the accurate input of licensing data. How does the tool ensure or verify the accuracy of the licensing information fed into it?

4. Response to Rapid Changes in ML Field: The ML field is rapidly evolving. How adaptable is ModelGo to the fast-paced changes in technology, especially in terms of accommodating new models, datasets, and associated licenses?

5. Global Applicability and Legal Jurisdictions: Does ModelGo take into account the variations in legal frameworks and copyright laws across different countries? How does it handle licensing issues that span multiple legal jurisdictions?

**Ethics Review Description:**

If ModelGo handles sensitive or proprietary data in its analysis of machine learning projects, it must ensure data privacy and security. This is particularly relevant if the tool accesses datasets or models that contain personal or confidential information.

**Ethics Review Flag:**

Yes

**Reviewer Confidence:**

3: The reviewer is confident but not certain that the evaluation is correct

**Scope:**

3: The work is somewhat relevant to the Web and to the track, and is of narrow interest to a sub-community

---

### Official Review · Reviewer_yuGm · 2023-11-23

**Novelty:** 6
**Technical Quality:** 5

**Review:**

The paper proposes ModelGo, a tool for auditing potential legal risks in machine learning (ML) projects. The authors argue that the traditional OSS license analysis cannot be directly extended to ML projects as they further include datasets and models which may be under different types of license. The authors also propose a taxonomy bridging AI activities and license language keywords. Utilizing their tool the authors analyze five diverse ML projects and generate assessment reports. Additionally, the authors also provide guidelines for minimizing license violation links.

The paper is well-written and easy to follow. I also feel the topic is timely and relevant given the current growth and proliferation of AI systems.

Often, the ML projects developed in the Academia are not properly organized in terms of licensing. I would like to know how difficult it is to deploy the developed ModelGo tool to a ML project. I would imagine that it would require manual effort to incorporate the missing information.

Also how do the authors ensure that the proposed taxonomy is comprehensive and encompasses all AI activities and more importantly when a new activity is proposed how easy or difficult it is to extend the tool to it.

I thank the authors for their response. I will stick with my original ratings.

**Questions:**

Please check the Review section.

**Reviewer Confidence:**

3: The reviewer is confident but not certain that the evaluation is correct

**Scope:**

3: The work is somewhat relevant to the Web and to the track, and is of narrow interest to a sub-community

---

### Official Review · Reviewer_B3Bx · 2023-11-24

**Novelty:** 6
**Technical Quality:** 7

**Review:**

This paper provides a potentially helpful tool to identify licensing contradictions or errors in machine learning projects. I am not a licensing or legal expert, but I very much believe this is an important topic, and one that machine learning practitioners should take much more seriously. I appreciate the push toward developing tools to assist those building these projects in making informed choices about what they can and should not do or use, and the ethics and appropriateness of the license attached to the resulting ML product.

The main drawbacks of this paper for me are the immense amount of legal/licensing jargon to wade through. I don't think there's a way around this, and the authors do provide helpful summary blocks. But as someone not seeped in legal jargon, I struggled to read this paper.

This doesn't mean it won't still be helpful to those building machine learning projects, it just makes it a challenging read.

EDIT: I think the authors addressed my concern in the comments, and, though I'm (still) not an expert in licensing details, I think it's really important to make this discussion central to ML work.

**Questions:**

I am curious (as I am not a legal expert), about other legal considerations. Like is transforming a text into vectors negate the original licence? That is, when you scramble the text so much, does that mean something? (I'm really showing my legal naivete here, but I have some memory that that form of transformation means something legally).  That is, are there other legal/license considerations here beyond just the actual combinations of licences themselves?

**Ethics Review Description:**

No human subjects involved - no ethical issues

**Reviewer Confidence:**

1: The reviewer's evaluation is an educated guess

**Scope:**

3: The work is somewhat relevant to the Web and to the track, and is of narrow interest to a sub-community

---

### Official Review · Reviewer_QMu7 · 2023-11-24

**Novelty:** 6
**Technical Quality:** 5

**Review:**

Summary: The paper studies the problem of understanding ML licensing in many different settings, such as when using OSS, pre-trained models, or various datasets. The authors explain the problem of verifying software licenses, especially when a model might have one license, but the data it is trained on might have a different license. They describe the ModelGo system, which checks for licensing conflicts and evaluate it on a series of case studies. The paper, as it's written, takes the perspective of a software developer who wants to know whether a set of tools with different licenses can be used together risk-free. On a technical level, this seems to be evaluating a network of dependencies for conflicts between licences.

Strengths:
- This is a very important problem, especially with the influx of LLMs and the legal questions around their use of copyrighted data. I'm excited to see a paper working to tackle this issue.
- I appreciate that the paper takes a close look at different popular datasets and models used and provides concrete, urgent examples of this problem (eg the Getty Images case). These examples make it all the more clear that this is an important area of research.
- Overall I found the paper well-organized and clear to read.

Weaknesses:
- I think the paper could be improved by incorporating more information about what legal scholars think about this area. For example, I would be interested to know how legal scholars are taxonomizing different components of ML projects. Adding a bit more related work or context in this area might help. I think this is particularly true in light of developments such as the guarantee that ChatGPT will handle copyright issues for any of its clients.

**Questions:**

How do legal scholars taxonomize different components of ML projects?

**Reviewer Confidence:**

3: The reviewer is confident but not certain that the evaluation is correct

**Scope:**

4: The work is relevant to the Web and to the track, and is of broad interest to the community

---

### Decision · Program_Chairs · 2024-01-22

**Decision:**

Accept (Oral)

**Comment:**

Our decision is to accept. Please see the AC's review below and improve the work considering that and the reviewers' feedback for cemera-ready submission.

"Reviewers unanimously found this to be a high quality paper tackling an important and timely problem. It would make a great addition to TheWebConf program."